# Agricultural Landscape Composition Linked with Acoustic Measures of Avian Diversity

**Adam P. Dixon \***, **Matthew E. Baker and Erle C. Ellis**

Geography & Environmental Systems, University of Maryland, Baltimore County, MD 21250, USA;
mbaker@umbc.edu (M.E.B.); ece@umbc.edu (E.C.E.)

\* Correspondence: adampdixon@umbc.edu; Tel.: +1-410-455-2002

**Abstract:** Measuring, monitoring, and managing biodiversity across agricultural regions depends on methods that can combine high-resolution mapping of landscape patterns with local biodiversity observations. This study explores the potential to monitor biodiversity in agricultural landscapes by linking high-resolution remote sensing with passive acoustic monitoring. Land cover maps produced using a small unmanned aerial system (UAS) and PlanetScope (PS) satellite imagery were used to investigate relationships between landscape patterns and an acoustically derived biodiversity index (vocalizing bird species richness) across 12 agricultural sample locations equipped with acoustic recorders in Iowa, USA during the 2018 growing season. Statistical assessment revealed a significant direct association between vocalizing bird richness and percent noncrop vegetation cover. High spatial resolution (1 m) UAS mapping produced stronger statistical associations than PS-based maps (3 m) for landscape composition metrics. Landscape configuration metrics (Shannon's diversity index, contagion, perimeter-area-ratio, and circumscribing circle index) were either cross-correlated with composition metrics or unusable owing to complete landscape homogeneity in some agricultural landscape samples. This study shows that high resolution mapping of noncrop vegetation cover can be linked with acoustic monitoring of unique bird vocalizations to provide a useful indicator of biodiversity in agricultural landscapes.

**Keywords:** croplands; anthromes; human dominated ecosystems; acoustic index; land cover; farmland; bioacoustics; working landscapes; conservation

## 1. Introduction

Agriculture now covers nearly 40% of Earth's terrestrial surface [1]. Together with forestry, settlements, and other infrastructures, agricultural land use has transformed more than three quarters of Earth's land into the heterogenous working landscapes of anthromes, within which many wild species sustain themselves in fragments of remnant, recovering and less-used habitats [2–4]. Tools for monitoring, conserving, and restoring biodiversity within these intensively used working landscapes are therefore increasingly in demand [5,6].

New remote sensing tools are creating unprecedented capacities to map and measure ecological form and functioning in agricultural landscapes [7–9], including unmanned aerial systems (UAS) deploying optical sensors capable of mapping spectral and structural patterns of vegetation at increasing spatial resolutions [10] and new forms of high temporal and spatial resolution satellite imaging [11]. Separately, or in association with imaging systems, passive monitoring systems, including acoustic recorders, radio tracking chips, and camera traps are emerging as low-cost tools for assessing the presence, abundance, and species richness of a variety biological taxa, including insects, birds, amphibians, and mammals [9,12–14]. Combining these technologies can enable powerful and detailed measurements of biodiversity across landscapes at lower costs than census-based approaches, through

empirical relationships between landscape and acoustic measures at sampled locations [9,15]. These techniques may then be used to monitor the success of multifunctional landscape management strategies aimed at improving agricultural production while conserving and restoring biodiversity and ecosystem functions [16].

Previous approaches to estimate biodiversity in relation to agricultural landscape structure have relied on labor-intensive and costly fieldwork to acquire data on both species and habitat characteristics [17,18]. The advent of low-cost passive acoustic monitoring and high resolution UAS and satellite imagery offers the potential to lower barriers to measurement by greatly reducing field costs [9]. Passive acoustic monitoring is increasingly being used to characterize wildlife populations, yielding results as good as or better than conventional approaches for estimating species occurrence [19]. Still, it remains unclear whether and how these passive monitoring techniques might be used effectively for this purpose in intensively used agricultural landscapes.

This study examines whether relationships between avian diversity and agricultural landscape patterns can be observed by combining low-cost acoustic monitoring with high-resolution remote sensing data from UAS and PlanetScope (PS) satellite borne optical sensors. To this end, a select set of methodologies was tested, including different strategies for mapping and assessing land cover composition and configuration, and for performing bioacoustic measurements across a sample of intensively managed agricultural landscapes in Iowa (Figures 1 and 2). Our primary goal was to assess whether vocalizing bird diversity derived from passive acoustic monitoring can be associated with differences in the composition and configuration of agricultural landscapes. Secondary goals were to compare different approaches to estimate avian diversity using passive acoustic monitoring and to assess how different imaging systems might alter the capacity to observe non-agricultural habitats which may be small or narrow within intensively used agricultural landscapes.

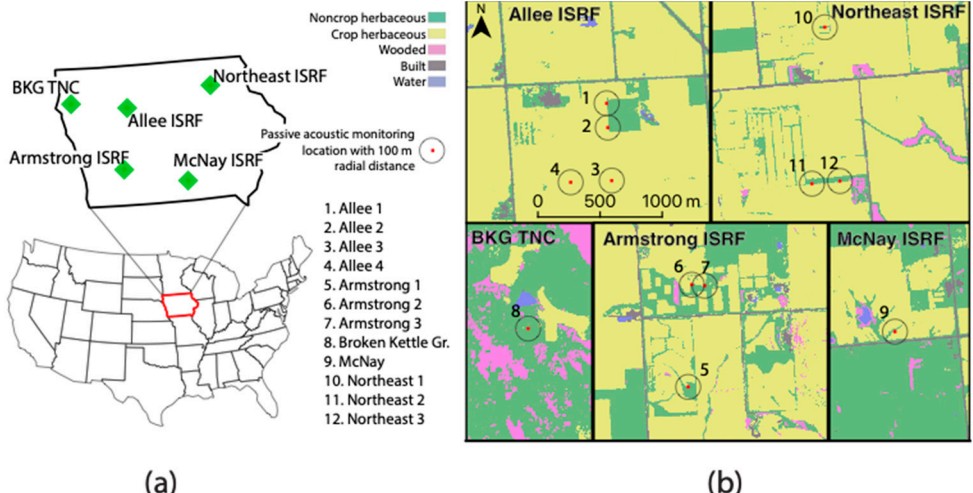

**Figure 1.** The study area (**a**) consisted of five sites across Iowa, USA. Passive acoustic monitoring locations (N = 12) are shown with land cover classification (**b**) in the background developed using PlanetScope (PS) imagery. Research and Demonstration Farms (ISRF) are managed through Iowa State University. Broken Kettle Grasslands (BKG) is a grassland preserve managed by The Nature Conservancy (TNC).

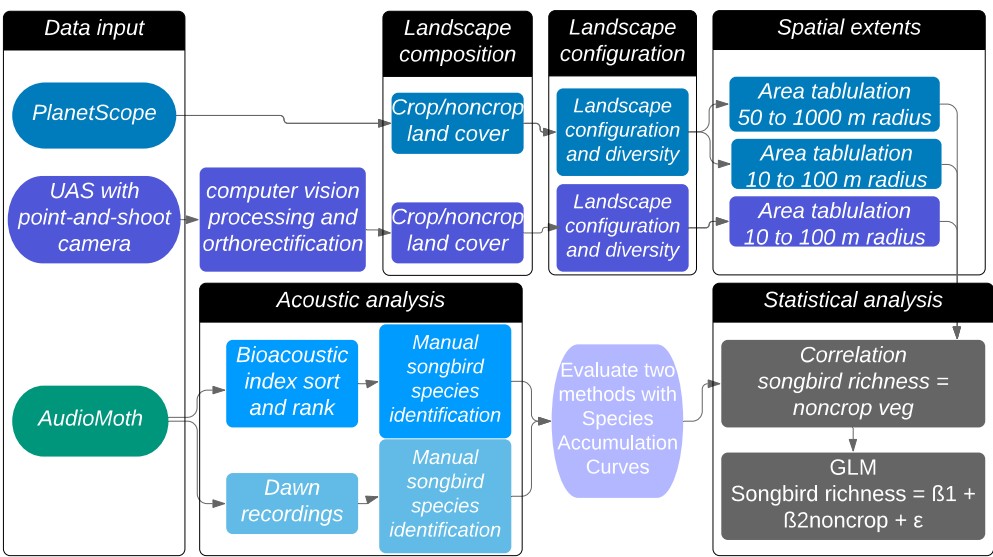

**Figure 2.** A flowchart representing the analysis steps.

## 2. Materials and Methods

### 2.1. Research Sites

Study areas are located in the Iowa Corn Belt [20]. Iowa was chosen because it is representative of intensive US Corn Belt agriculture. Study sites were selected through outreach with Iowa State University Research and Demonstration Farms (ISRF) and a Nature Conservancy grassland preserve. Passive acoustic monitoring locations were then identified based on availability through agreements with property managers and on degree of variation in total proportion of noncrop vegetation cover in the USDA Cropland Data Layer (CDL) within 1 km [21], providing a low resolution assessment of potential monitoring locations.

Study areas and sample sites were selected to provide broad ranging and representative observations of common variations in the structure and composition of intensive agricultural landscapes to serve as a platform for investigating new tools for assessing landscapes and acoustic measures of avian diversity in a single season of field research. Once suitable and accessible study areas were identified, wooded areas and water were avoided in selecting sample sites.

### 2.2. Acoustic Data Collection

Acoustic recorders were placed along field margins in collaboration with farm and site managers to avoid conflict with management operations. Three sites had more than one recorder due to travel logistics and access constraints. When more than one recorder was placed at a site, they were placed at least 200 m apart, with the exception of one site where recorders were placed 100 m apart (Armstrong ISRF) due to onsite space constraints. The 200 m distance is often used when establishing traditional avian point counts to avoid overlap of local landscapes [22].

Acoustic recordings were collected with an AudioMoth (v1.0), a small, inexpensive, and lightweight recording device [13] (Figure S1). Installation procedures were designed to capture sounds near the top of the herbaceous canopy. The acoustic monitors were placed within a simple plastic bag with locking zip seal and fastened one meter above ground on either a wooden stake or a fencepost with a bungee cord facing northwards. (North was chosen for no other reason than to pick a consistent direction.) The entire set of recordings ranged between sites from 6 to 41 days of recording from late June to mid-August 2018. This wide range of recording dates was due to project funding and recorder availability constraints. Five recorders were initially available, with seven more available only in

August 2018. All units were programmed to have a sample rate of 48 kHz for 1 min every 10 min to address data storage limitations to enable long term acoustic sampling.

### 2.3. Assessing Avian Diversity from Acoustic Recordings

The primary spatial extent for relating acoustic monitoring and landscape analysis was chosen in relation to the distance from the acoustic recorder within which avian vocalizations were detectable. Avian diversity was quantified by counting unique bird species vocalizations from acoustic recordings processed using two approaches aimed at filtering optimal recordings for bird identification: Bioacoustic Index sampling (BIO) and dawn sampling.

#### 2.3.1. Acoustic Detection Extent

The spatial distance from the acoustic recorder within which avian vocalizations were detectable was assessed empirically. An AudioMoth unit was set up according to study site conditions. Amplitude attenuation was measured at distances ranging from 1 to 250 m. A red-winged blackbird (*Agelaius phoeniceus*) song vocalization was chosen because of its ubiquity across most sample sites, and because the maximum sound pressure level (SPL) at 1 m distance has been estimated at 91 dB [23]. The playback of a red-winged blackbird song downloaded from the xeno-canto database [24], was set to 91 dB using a sound meter (accuracy ± 1.5 dB) placed 1 m from the playback speaker.

The acoustic recorder was positioned facing slightly northeast (instead of direct north like study site procedures) to achieve the longest possible straight-line distance in the testing area. At the time of testing, wind conditions averaged 2.5 m per second from the west-southwest with some wind gusts up to approximately 7 m per second (~15 mph). Playback of the vocalization was repeated at distances ranging from 1 to 250 m. A band-pass filter was applied from 2 to 8 kHz to remove extraneous sounds, and a red-wing blackbird vocalization was clipped and saved from each recording distance to measure the signal-to noise ratio (SNR) and maximum SPL. An SNR of 1 indicates that the strength of the signal does not exceed background noise and will be either faint or undetectable. The SPL was measured to note the difference between expected and observed attenuation.

SNR was consistently above 1 within 100 m of the recording device, though it varied greatly owing with distance to variable wind conditions (Figure 3a). The SPL followed theoretical attenuation by spherical spreading measured with amplitude level $l_{ref}$, distance from the source $d$, and distance from the source of the reference level $d_{ref}$ (equation in Figure 3b) [25]. Based on these observations, a 100 m spatial extent from the sample point was adopted as the primary spatial extent to test for relationships between landscape and acoustic measures.

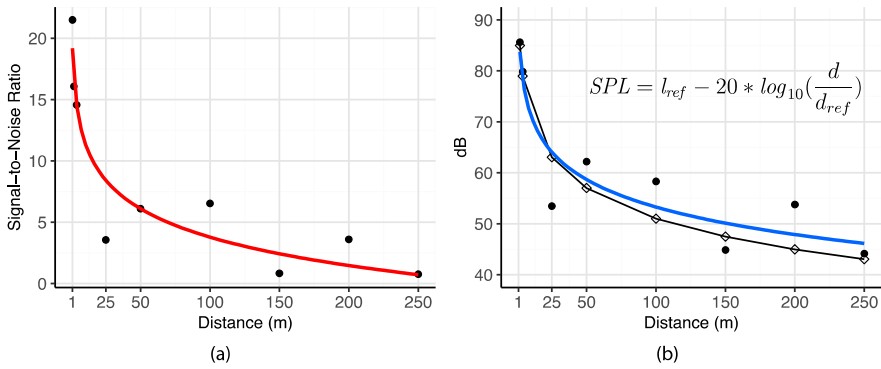

(a)

(b)

**Figure 3.** Attenuation of bird vocalization detectability over distance from sensor. Plot (**a**) is the measured signal-to-noise ratio (SNR). Plot (**b**) shows the measured sound pressure level (blue line fitted by logarithmic curve) versus theoretical attenuation (equation provided) of a 91 dB sound (black line) by spherical spreading.

### 2.3.2. Bioacoustic Index Sampling

The Bioacoustic Index (BIO) value of each recording was used to rank the full set of recordings (Supplementary Materials). BIO was developed by Boelman et al. (2007) [26] and is calculated by generating values for amplitude and frequency and then quantifying the area under the curve of an amplitude-frequency plot. The area values represent a function of the sound level and the number of frequency bands within the 2–10 kHz frequency range [26]. Instead of limiting the recordings at each sample site by a date range or period, the entire set of recordings were evaluated for a BIO value.

All recordings were ranked and sorted based on the resulting index value. The 100 acoustic recordings collected at each site with the highest values were selected for manual identification of vocalizing bird species. Next, each acoustic recording was individually reviewed for unique vocalizing bird calls and songs in Raven Pro software (Cornell 2012), a graphical acoustic analysis software platform that allows for playback and spectrogram annotation. All vocalizing birds were identified by A.D.

### 2.3.3. Dawn Sampling

In addition to the BIO assisted sampling approach, a dawn sampling method was developed. Dawn sampling has been shown to exceed the ability of traditional point count methods by selecting a set of recordings from at least 3 h following dawn and manually identifying species [19]. Since the acoustic recorders were programmed to record for 1 min every 10 min, all recordings following sunrise (approximately 06:00) until approximately 09:30 local time for six days were used, generating 120 recordings for evaluation per site. Effort was made to keep dawn recordings as close in date range as possible. Ten of the sites had recordings evaluated between the dates of 9 to 15 August 2018. Recordings from the two other sites (Broken Kettle Grasslands, McNay ISRF) were selected to align as much as possible with these sites. Broken Kettle Grassland recordings were from a noncontinuous set of days in July 2018 (11–13 July; 21–24 July) due to a set of recorder malfunctions. Recordings for McNay ISRF were from 3 to 8 August 2018.

### 2.3.4. Comparing Sampling Approaches Using Species Accumulation Curves (SACs)

To evaluate the effectiveness of each approach, SACs were produced to determine the limits of sampling effort required across sites. SACs were developed using the Vegan package in R using rarefaction (with 100 permutations in recording sample order) [27] (R version 3.6.1; The R Foundation for Statistical Computing 2019). The rate of identification of unique vocalizations from the last 25 rarefaction estimates of each site were then used to indicate effort required to add one additional species.

### 2.4. UAS and Satellite Imagery Data Collection

The UAS was a DJI Matrice 100 deployed with a NIR modified Canon Powershot Elph 190 converted by LDP LCC, Carlstadt, NJ, USA (maxmax.com). The Canon Hack Development Kit (CHDK) was installed on the camera setting the intervalometer to capture an image every second. The gray/white balance was calibrated before each flight with a gray/white reference board. Image acquisition flights took place in August 2018. Flight plans were established to capture images with 50% overlap, with a flight altitude of 70 m above ground. Imagery was postprocessed in Agisoft Photoscan software (version 2.7) and mosaicked with three spectral bands (B, G, NIR). A projected coordinate system (UTM 15N) was applied to the ungeoreferenced orthophoto with QGIS Georeferencer and using ground control points taken during the UAS flight and an online map reference (Google earth) [28]. Resolution of imagery produced was approximately 2 cm. Imagery was resampled to 1 m using median values to increase processing speed, reduce the range of pixel values used to distinguish land cover classes, and to avoid classification of shadows.

PS imagery is a 3.7 m ground sample distance product resampled to 3 m with approximately five-day temporal resolution [11]. PS imagery was downloaded for each study site as a four-band (B, G, R, NIR) product with surface reflectance correction [29,30]. Cloud-free scenes were identified within the growing season between early June and early September of 2018 at least one week apart to allow for some change in the phenology of the vegetation. Each cloud free scene was merged into a time-series stack of imagery. The total number of image dates within the time-series stacks ranged from five to eight. The PS imagery was mosaicked while maintaining separate spectral bands. This resulted in time-series stacks for each site with between 20 and 32 spectral bands. All post-processing was completed using GDAL utilities in QGIS 3.1 [31].

### 2.5. Land Cover Composition and Configuration

The 1 m resolution UAS imagery and 3 m PS imagery were used to produce separate land cover maps. A simple five class land cover classification system was used to differentiate distinct features in agricultural landscapes: herbaceous noncrop, herbaceous crop, woody, built, and water. The herbaceous noncrop class represents all non-woody vegetation that is not a row crop. The herbaceous crop includes all row crops, which at our study sites are corn and soybeans. The woody class represents all trees and shrubs. The built class represents all infrastructure including roads. The water class represents all open surface water. After classification, woody cover was combined with herbaceous noncrop cover to create a noncrop vegetation class. Noncrop vegetation is therefore defined as the combination of all herbaceous noncrop and woody vegetation cover, some amount of which might include woody crops, though these are rare in the Iowa Corn Belt. Ground truth data were collected using geo-tagged smartphone photographs and were accompanied by descriptions of plant species observed and then cross-referenced with an online map reference (Google earth) to guide delineation of training polygons for both UAS and PS imagery. Seventy percent of the pixels within the training polygons were randomly extracted and applied to the training algorithm, with the remaining 30 percent used to test model performance. The linear support vector machine (SVM) classifier within the Caret package in R [32] was used to map land cover classes, which were then applied to the imagery using the Raster package [33]. The SVM was parametrized using a linear kernel with 5 K-fold cross validation. SVM is a widely used classifier in grassland remote sensing because of its ability to detect variation of spectral reflectance in homogenous vegetation patches with improved performance in grassland systems with increasing temporal resolution [34,35].

### 2.5.1. Configuration Metrics

Several measures of landscape configuration were calculated (Table 1). Perimeter-area ratio (PAR) measures differences in noncrop vegetation patch shape [36]. Circumscribing circle index (CIRCLE) measures overall noncrop vegetation patch elongation, with narrow and elongated patches having a high CIRCLE index [36]. Only noncrop land cover types were used in the development of CIRCLE and PAR index values, thereby excluding crop, built, and water land cover classes from influencing results. Shannon's diversity index (SDI) and the Contagion index were calculated using the full five class land cover classification. SDI is a proportional representation of land cover types. Contagion is a measure of the isolation and proximity of land cover classes with low values indicating that land cover classes are in many patches and dispersed, and high values indicating that land cover is mainly in large contiguous patches [36]. Configurational measurements were completed using the LandscapeMetrics package in R [37] and tabulated for all spatial extents.

**Table 1.** List of landscape indicators tested for association with vocalizing bird richness.

| | |
|---|---|
| Noncrop vegetation | Potential habitat area, as proportion (%) of herbaceous noncrop and woody vegetation (composition) |
| Shannon's Diversity Index (SDI) | An index value measuring relative proportional representativeness of all land cover types (configuration) |
| Contagion | An index value measuring patch isolation and proximity of all land cover types (configuration) |
| Perimeter Area Ratio (PAR) | Ratio of total perimeter of noncrop vegetation patches divided by the total area (configuration) |
| CIRCLE | Circumscribing circle index to measure linearity of noncrop vegetation patches using area and radius (configuration) |

### 2.5.2. Spatial Extents

To test for scale-related changes in computing landscape indicators and relating these to biodiversity measures, nested circular spatial extents were created at increasing radial distances around each passive acoustic monitoring location to assess noncrop vegetation proportions and landscape configuration metrics (Figure 4; every 10 m up to 100 m, and every 50 m up to 1000 m). Total areas of noncrop vegetation within each radial spatial extent were then converted to a proportion of the total area of the circular extent (noncrop vegetation cover percent). SDI and Contagion were global measures, while CIRCLE and PAR were mean values within each spatial extent. All data processing was completed in R with the raster and sf packages [38,39].

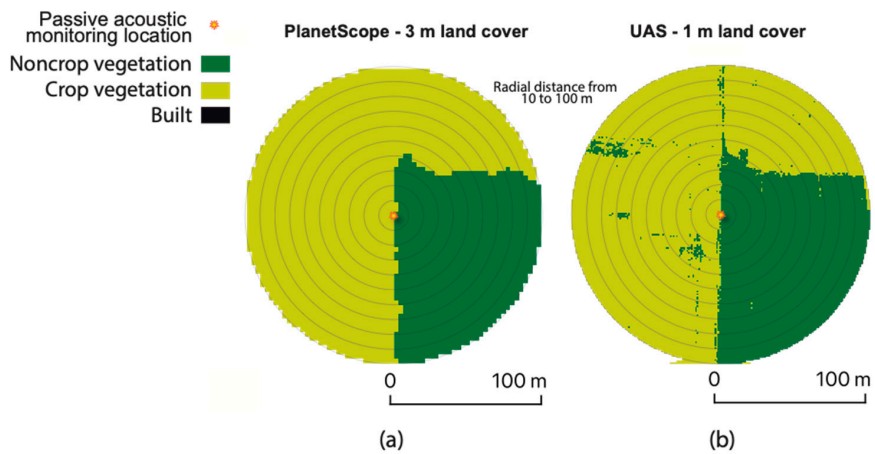

**Figure 4.** Land cover characteristics from (**a**) PlanetScope (PS) and (**b**) UAS imagery were analyzed within 100 m of the passive acoustic monitoring location.

### 2.6. Statistical Modeling

To assess the effect of spatial resolution and extent, Pearson's product-moment correlation was used to evaluate noncrop vegetation proportions measured at each spatial extent and vocalizing bird richness [40]. Multi-species studies have utilized a nested spatial extent landscape sample design [40] to identify both the scale at which detections are strongest statistically and to rule out smaller and larger spatial extents [41]. Measures of richness among taxonomically related groups are expected to respond to a similar range of spatial extents [17,42]. Data exploration through correlation analysis, collinearity, and normality tests were used to check for outliers, ensure statistical assumptions were met, and provide a preliminary understanding of relationships between variables.

A predictive model was then developed using only the target 100 m radial extent, chosen based on detection extent of the acoustic recorder (Figure 3). Generalized Linear Models (GLM) with a Poisson distribution examined the strength of the relationship using both UAS and PS land cover data.

Poisson regression was used because the relationships observed were not expected to be linear due to agricultural landscape complexity theory [43,44], and were based on count data for the response variable. GLMs were assessed for explanatory power using explained deviance, which is equivalent to an $R^2$, and is measured using the residual deviance and that of a perfectly fit model [45]. Model diagnostic tools including Cook's distance, linearity, and normality were used to check for outlier influence and ensure GLM assumptions were met.

## 3. Results

### 3.1. Acoustic Data Analysis

A total of 55 unique bird vocalization patterns were identified throughout all sites using the BIO and Dawn methods of species identification (Table S1). Of these, 45 were identified to species and 10 were not. Unidentified unique vocalization patterns were counted as additional species but marked as unknown. Species lists are provided in the Supplementary Materials (Tables S2 and S3). The BIO sorted rank method identified between 3 and 16 unique vocalizations at each site, while the dawn method identified between 4 and 22 (Figure 5). The number of recordings needed to add one species was approximately 38 and 39, for the BIO and Dawn methods respectively, suggesting similar outcomes regardless of method. Results of both methods were correlated (Pearson's $R = 0.72$). The Dawn method was chosen over BIO for further research because it was taken from a more consistent timeframe than the BIO samples, and offered greater discrimination among sites as is seen in the spread of number of species identified by BIO and Dawn in Figure 5a,b. Individual site SACs are provided in Figure S2 with error bars that demonstrate the range in species number identified given listening effort with random reordering of recording samples.

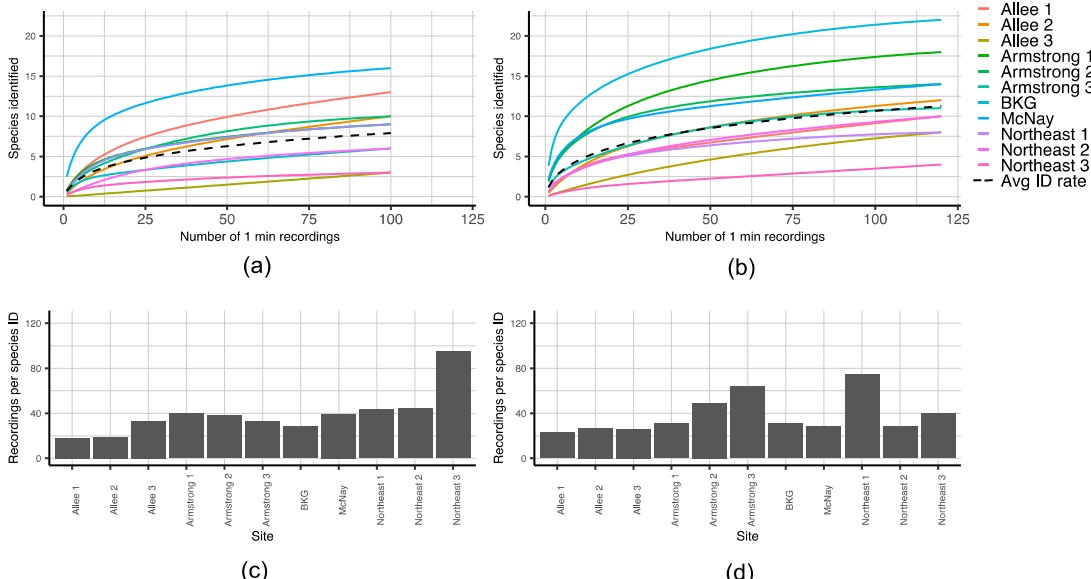

**Figure 5.** Number of species detected in relation to number of 1-min recordings, illustrating species detection approaching saturation within 100 to 125 recordings. The Bioacoustic Index sorted rank method (**a**) led to a narrower range in species identification than the Dawn recordings method (**b**), which was chosen for use in further analysis. Barplots show the rate of identification displayed as total number of additional recordings needed to identify an additional species at each site for (**c**) the Bioacoustic Index rank and sort, and (**d**) the dawn recordings method. BKG = Broken Kettle Grasslands.

### 3.2. Land Cover Mapping

Land cover map classification accuracy was >88% for both UAS and PS imagery (Figures 6 and 7). Average total accuracy and Kappa for UAS maps were 89% and 0.8, respectively, and even higher for

PS maps; 99% and 0.9 (Tables S4 and S5). The land cover classes most commonly misclassified in UAS data were noncrop herbaceous vegetation and herbaceous crops, followed by errors of commission where woody vegetation was classified as herbaceous crops. No statistically significant errors were identified in the classification of PS imagery.

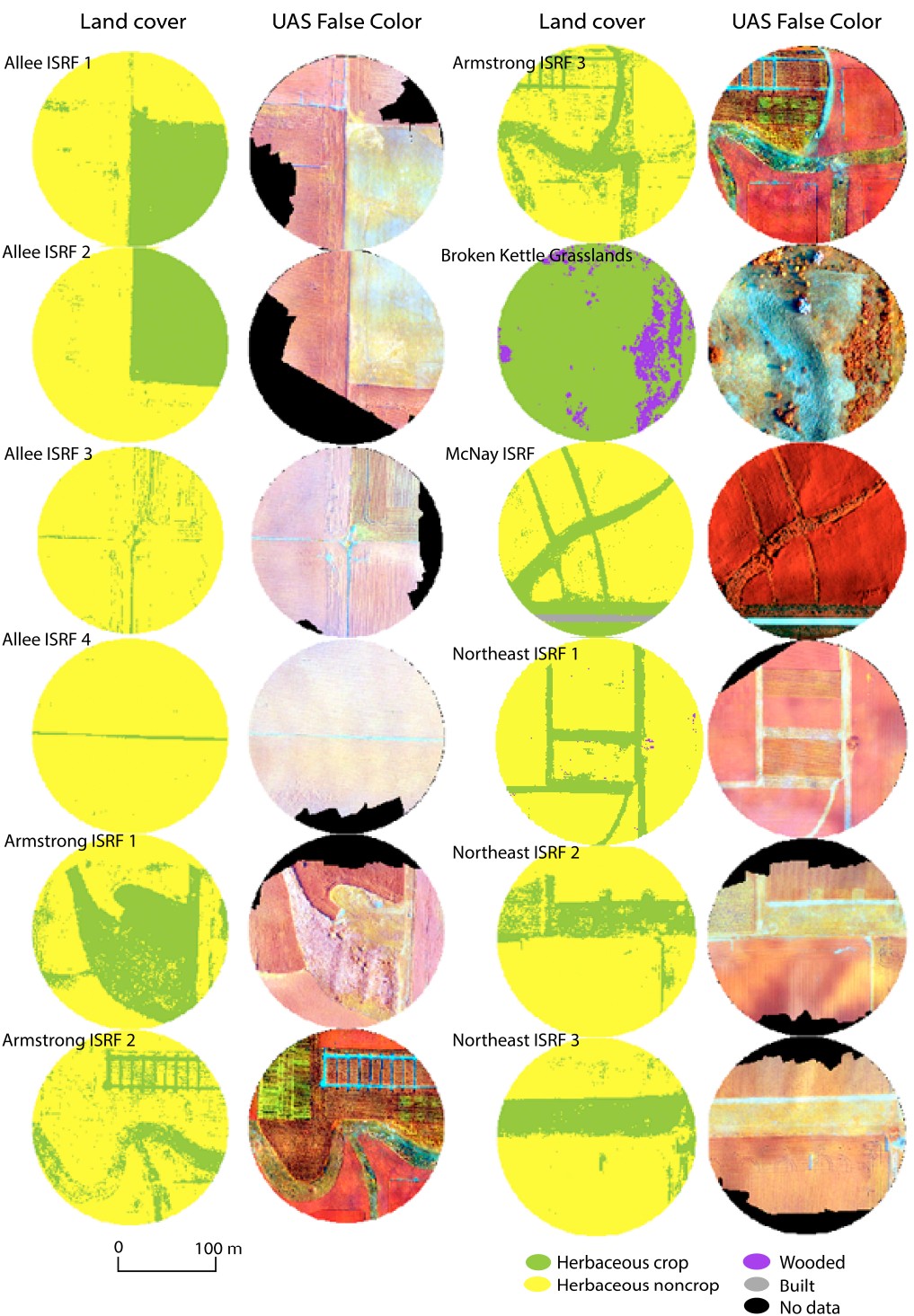

**Figure 6.** UAS land cover maps and false color images. No data values in raw images were the result of inadequate photographs for computer vision mosaicking. These areas were compared with the 2018 USDA Copland Data Layer [30] and manually reclassified as herbaceous noncrop.

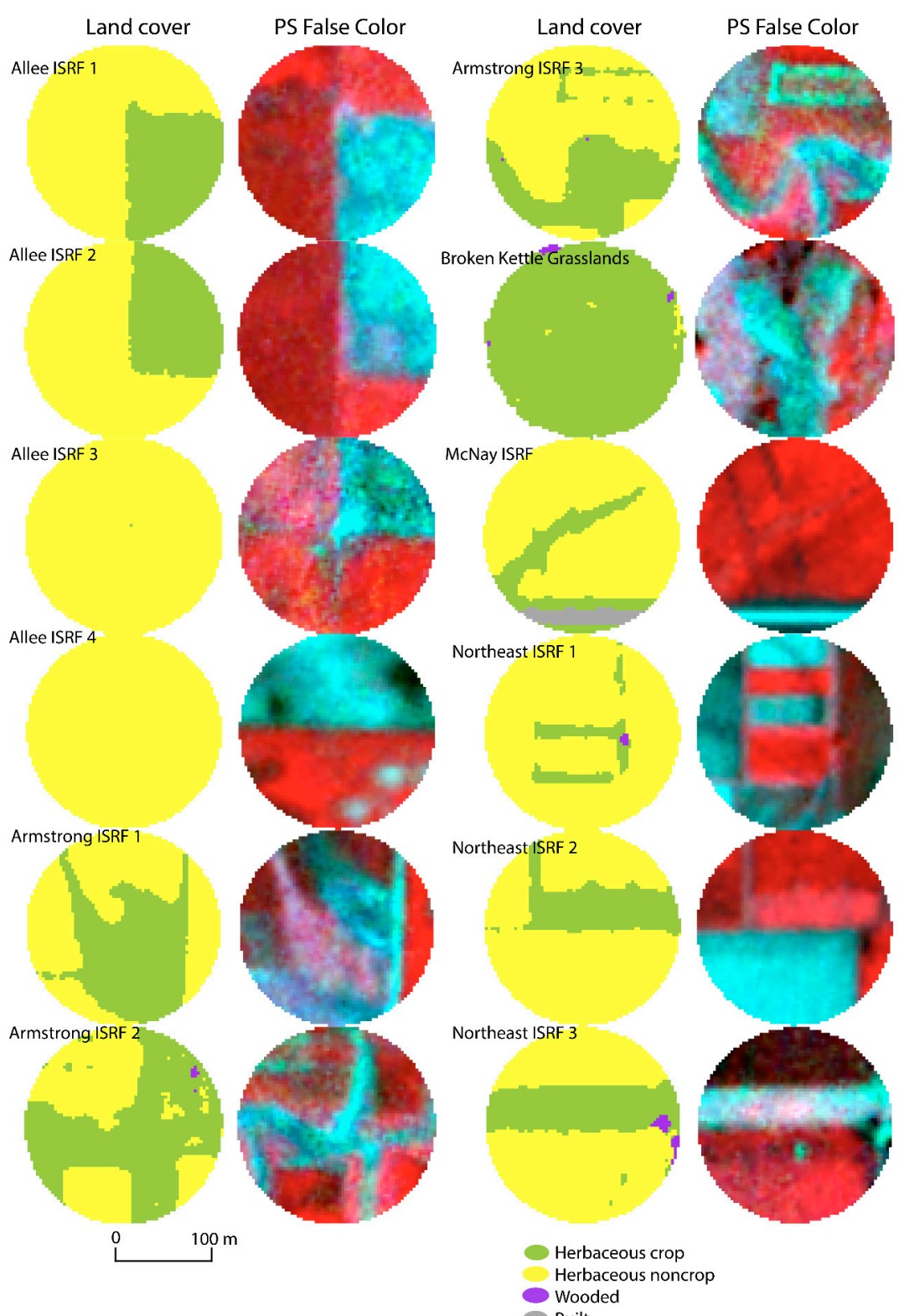

**Figure 7.** PlanetScope (PS) land cover maps and false color images.

### 3.3. Spatial Extents and Resolutions

Noncrop vegetation cover proportion, the key landscape composition indicator of this study, was highly spatially autocorrelated when computed across tested spatial extents ranging from 10 to 100 m (Figures S3 and S4). A 100 m radial extent was therefore confirmed as the standard spatial extent to

test for statistical relationships between landscape and acoustic indicators, as this was the acoustic detection extent of the acoustic recorder used in this study.

The noncrop vegetation proportion within 100 m of acoustic recorders was tested for a normal distribution across sample sites, and Broken Kettle Grasslands (BKG) was removed from subsequent analyses as an outlier, reducing the total number of landscape/acoustic samples used from N = 12 to N = 11. It is notable that BKG simultaneously had both 100% noncrop vegetation cover within 100 m of the acoustic recorder and the highest number of species identified at any site (22). We reported BKG because it was situated among row crop agriculture just outside of the 100 m radius of the acoustic recorder (Figure 1b).

The spatial resolution of land cover data had statistically significant effects on indicators of landscape configuration, based on comparisons of PS and UAS data (Figure 8). This was largely the result of narrow strips of noncrop vegetation (<2 m) that were undetectable using PS 3 m resolution imagery, yielding maps with zero noncrop vegetation and 100% crop vegetation cover in two sites. SDI and Contagion indices require maps with areas of more than one land cover class and were therefore excluded from further analysis. Landscape linearity metrics, PAR and CIRCLE, could be computed, but their distributions and magnitudes were very different between PS and UAS data and highly correlated with noncrop vegetation proportion, making these indicators both redundant and lowering confidence in the interpretation of results (Figure 8). These landscape configuration metrics were therefore eliminated from further statistical assessment of linkages with biodiversity indicators.

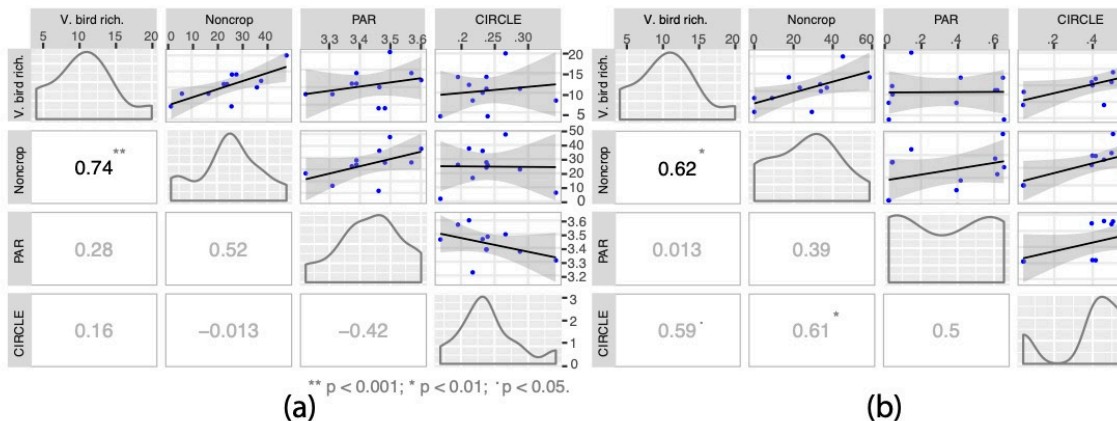

**Figure 8.** Correlations, data distributions, and scatterplots between vocalizing bird richness and landscape indicators, illustrating greater correlation for the noncrop indicator and cross-correlation of other indicators using both (**a**) UAS and (**b**) PlanetScope (PS) land cover data with outlier BKG removed (N = 11). Density plots are in the diagonal positions. Scatterplots on the upper right side include the black line fitted with a linear model gray shading as 95% confidence intervals. Labels are: V. bird rich.—vocalizing bird richness; Noncrop—proportion (%) of noncrop vegetation; PAR—perimeter area ratio; CIRCLE—circumscribing circle index.

### 3.4. Statistical Assessment of Landscape Indicators in Relation to Vocalizing Bird Richness

The landscape composition measure, noncrop vegetation proportion within a 100 m radius of an acoustic recorder, was used as an independent variable in testing for associations with vocalizing bird richness using GLM (Table 2, Figure 9). Again, in model diagnostics, Broken Kettle Grasslands (BKG) was identified as improperly influencing results based on Cook's distance and was dropped from the analysis, leaving a sample size of N = 11. Two Poisson regression models were estimated using the UAS and PS land cover data (Figure 9). UAS estimates of noncrop vegetation proportion explained 54% of the variation in vocalizing bird richness ($p < 0.001$; Table 2), while PS estimates explained only 38% of this variation ($p < 0.01$).

**Table 2.** GLM results indicating highly significant relationship between noncrop proportion and vocalizing bird richness for both UAS and PS landscape maps. Standard error is reported in parentheses.

|  | **UAS** |  | **PlanetScope** |  |
|---|---|---|---|---|
| (Intercept) | 1.68(0.24) | *** | 1.93(0.19) | *** |
| UAS Noncrop | 0.03(0.01) | *** |  |  |
| PS Noncrop |  |  | 0.01(0.01) | ** |
| N | 11 |  | 11 |  |
| AICc | 60.47 |  | 63.82 |  |
| Explained Dev. | 53.59 |  | 37.66 |  |

*** $p < 0.001$; ** $p < 0.01$.

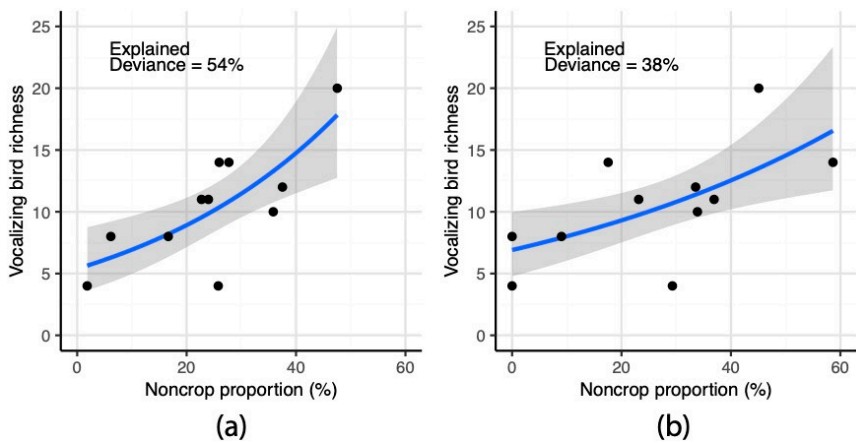

**Figure 9.** GLM results using a 100 m radial spatial extent to measure noncrop vegetation proportion for the UAS (**a**) and PlanetScope (PS) (**b**) data. The blue line indicates the fitted line, while gray shading indicates 95% confidence intervals.

## 4. Discussion

Results from this study demonstrate that measures of vocalizing bird diversity derived from acoustic recordings can be predicted from measures of agricultural landscape composition. High-spatial resolution UAS-derived land cover maps were especially robust in these predictions, explaining 54% of observed variation in vocalizing bird richness across 11 agricultural landscape sample sites when testing using GLM (Figure 9). PS-derived land cover maps also produced statistically significant predictions of vocalizing bird richness, but were less robust, explaining 38% of variation across sample sites. As seen in this and other studies [46], landscape homogeneity can be extreme in intensive agricultural regions, leaving only very fine-scale features of non-crop vegetation, such as field strips and margins with widths <1 m potentially serving as wildlife habitats. It is therefore unsurprising that high spatial resolution imagery is especially useful in mapping potential wildlife habitats in these regions. Still, a rough estimate of the extra time required to map landscapes with our UAS platform was about 15 hours per site, not including transport time (UAS flights may complement installation of passive acoustic monitors), and the UAS platform required an estimated US $4,000 of investment, not including computer vision software. When comparing the much higher time and financial costs associated with UAS deployment and data processing in relation to PS imagery, this difference of 16% in explained variation seems less significant.

There were several UAS data points (Armstrong 2, Armstrong 3, McNay, Northeast 2, Northeast 3) where vocalizing bird richness ranged from 4 to 14 while noncrop vegetation proportion remained approximately the same (about 25%; Figure 9a); a clear indication that unobserved factors may be influencing results. For example, management actions such as mowing or pesticide application at a

site may be more influential to vocalizing bird richness than the total proportion of habitat available. This may also be related to the use of a highly simplified land cover classification which was used to reduce complexity in the analysis, allowing our focus on the central objective of the study. However, the management of farmland mosaics is diverse, even across the intensively managed agricultural regions of Iowa, and this influences the degree to which landscape elements, such as hedgerows and linear features, can provide functional habitat for wildlife [47,48].

Though the sites included in this study were distributed across a broad slice of Iowa farmland geography, they were limited to a relatively small sample chosen in part for their accessibility, to enable rapid field assessment of new techniques in a single season. To test for geographical biases, sample locations were examined for north and south groupings, latitudinal placement, and distance between each acoustic recording location, yielding evidence for geographical effects (Table S7). Including these effects in GLM analysis did not lead to substantial improvements in model performance and were therefore not studied further here. However, these ecoregional effects could be important sources of explanatory power that may be revealed in future efforts using larger and more widely distributed samples. Another important consideration is that our study occurred late in the summer and in a single season. While we would expect the same community of resident breeding birds throughout the growing season, we acknowledge that results may have been different if acoustic activity was sampled a few months earlier when birds are establishing territory and pair-bonding. We look forward to future studies using long duration passive acoustic monitoring across multiple seasons which are able to determine the precise effects of seasonality and species detection.

Vocalizing bird richness, derived from manual counts of unique species calls from acoustic recordings, proved to be an easily computed indicator of avian diversity. Though it is an incomplete measure that counts only birds with vocalizations with long durations, fails to take into account species abundance or functional traits, and does not incorporate differences in range, dispersal, or turnover among species counted [41,49], we propose that it can serve as a useful and pragmatic metric for assessing avian diversity variations across agricultural landscapes that can be widely reproduced using inexpensive acoustic recordings. Additional studies will be needed to verify our results as they are not readily compared with existing datasets like the Breeding Bird Survey and eBird which operate at larger scales and include woodland and water birds that rarely occur in the cropping systems we analyzed [50,51].

Passive acoustic monitors are well suited for studies occurring across large geographic areas [52]. For the current study, we estimate that passive acoustic monitoring took about as much time (two site visits to install and collect recorders) as conducting two bird point count estimates at a site, but was able to collect observations over multiple (up to 41) days. Studies utilizing point count surveys have suggested a minimum of four visits on different days are required to monitor cropland birds at agricultural sites [53]. Using acoustic monitoring, we estimate that manually identifying all species in one minute of recorded audio requires about two minutes. Using dawn sampling, the 12 sites of our study produced 1440 sample minutes, requiring about 48 h of manual assessment. So, site point counts would require at least twice as much site travel time as acoustic monitoring, against the additional time for interpretation of audio recordings. Fortunately, bird identification from audio recordings may soon be fully automated, eliminating most of the time and labor required [54–57].

Previous studies on Iowa birds found that that measures of farmland landscape composition were often stronger predictors of avian biodiversity than configurational measures [58–61]. Our results are similar to these and other prior studies demonstrating the greater utility of measures of landscape composition, such as habitat area and proportion, over measures of landscape configuration, in predicting biodiversity and other ecological indicators [62,63]. Previous studies have also demonstrated a high dependence of configurational measures on the spatial resolution of the data used to compute them [64,65], with higher spatial resolutions resulting in higher variability of configurational measures [66,67]. Though the configuration measures in our study did correlate with the response variable, they provided no greater correlations than did composition.

Combining acoustic monitoring with fine-scale land cover mapping highlights the importance of small habitat complexes in places typically overlooked for their conservation potential in the industrial agricultural landscapes of the US Great Plains. Moreover, using birds as an indicator of biodiversity and other ecological patterns and processes is a classic research approach that can be expanded across terrestrial systems thanks to the availability of low-cost acoustic recorders. Use cases for this research approach are numerous in agricultural landscapes and range from evaluating the effectiveness of conservation interventions, such as the Conservation Reserve Program, to assessing the effects of farm production practices such as pesticide and fertilizer application. Outside of agroecosystems, passive acoustic monitoring provides numerous opportunities to monitor ecological change [68].

The new operational capabilities provided by combining high resolution UAS or PS imagery and passive acoustic monitoring open up a number of new research questions. For instance, variations within crop and noncrop vegetation compositional heterogeneity may also influence biodiversity in agricultural landscapes [43,44]. Future research may also investigate how species and/or species groups respond to agricultural landscape patterns, including the functional quality of noncrop vegetation; these questions remain unexamined using the methods and tools of this study. Nevertheless, linking acoustic measures of vocalizing bird richness with land cover mapping can help to understand avian responses to landscape composition, potentially enabling widespread monitoring of the biodiversity consequences of fine-scale changes in the management of agricultural landscapes.

## 5. Conclusions

This study prototypes the use of low-cost acoustic recorders and high resolution remotely sensed imagery to predict avian diversity across agricultural landscapes. Our findings of a strong statistical association of vocalizing bird richness with noncrop vegetation cover confirm prior research demonstrating strong linkages between landscape composition and biodiversity. Using higher spatial resolution UAS imagery to map land cover led to more robust statistical linkages between landscape composition and avian richness, owing to a greater ability to resolve the fine-scale scale features of noncrop vegetation often present in intensively management agricultural landscapes. However, PS imagery, which was far less costly, was also capable of measuring this linkage. These new and inexpensive methods for widespread measurement and monitoring of variations and changes in farmland landscapes and avian diversity offer new opportunities not only for scientists, but may also to enable farmers, ranchers, and advocates of environmental quality to participate directly in monitoring, understanding and managing the ecology of agricultural landscapes.

**Supplementary Materials:** The following are available online at http://www.mdpi.com/2073-445X/9/5/145/s1. Figure S1: Photographs of AudioMoth passive acoustic monitors deployed in the field, Figure S2: Individual species area curves using for each acoustic recording sampling method with error bars showing standard deviation from 100 permutations to compute the rarefaction, Figure S3: Pearson's correlation at each radial buffer distance for PlanetScope data from 50 to 1000 m. Error bars show 95% confidence intervals based on Fisher's r to z transformation, Figure S4: Noncrop vegetation proportion change by analysis extent using UAS data, Table S1: Noncrop vegetation proportions at a 100 m radius from sample site for UAS and PlanetScope (PS) data. Also included are songbird richness counts using the dawn sample approach and the bioacoustic index (BIO) sorted rank approach, Table S2: Vocalizing bird species codes for each site using the dawn recording sampling approach, Table S3: Species names and codes, Table S4: Confusion matrix for all UAS land cover classifications, Table S5: Confusion matrix for all UAS land cover classifications, Table S6: Proportion of UAS data with No data values These areas occurred due to limited coverage by photographs taken aboard the UAS. Areas were compared with the USDA Cropland Data Layer to confirm reclassification into crop vegetation, Table S7: GLMs examining the impact of geographic bias in the dataset.

**Author Contributions:** Conceptualization, A.P.D. and E.C.E. Methodology, A.P.D.; Formal analysis, A.P.D., M.E.B., E.C.E.; Writing, A.P.D. All authors have read and agreed to the published version of the manuscript.

**Funding:** This research was partially funded by the National Science Foundation, Ecosynth: An Advanced Open-Source 3D Toolkit for Forest Ecology, Award number 1147089.

**Acknowledgments:** We would like to thank Mark Honeyman, Lyle Rossiter, Ken Pecinovsky, Logan Wallace, and Dallas Maxwell of Iowa State University Research and Demonstration Farms and James Baker and Scott Moats of The Nature Conservancy in Iowa for facilitating research. We thank the anonymous reviewers for quality feedback. We acknowledge the Planet Education and Research Program for no-cost access to imagery used in this study. We also thank Ciara Hovis and Dorothy Borowy for providing early reviews of the manuscript.

**Conflicts of Interest:** The authors declare no conflict of interest.

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
