# Peer review of "Agricultural Landscape Composition Linked with Acoustic Measures of Avian Diversity"

_land, doi:10.3390/land9050145_

Round 1
Reviewer 1 Report
I have reviewed the manuscript Land_779378. Knowing the manuscript from previous reviews in the Remote Sensing journal, I clearly recognize several improvements, but still think that the manuscript can be improved. Some of the comments and questions indicated in the previous review are not answered yet. Because pf this, I consider that there are some details that should be revised in order to improve the manuscript. My comments are detailed below:
- The study compares two types of images with very high spatial and temporal resolutions but, are these high resolutions necessary? To classify herbaceous noncrop, herbaceous crop, woody (that merged to herbaceous noncrop to create a single noncrop vegetation category), built, and water, it is necessary a very high spatial resolution?. The very high spatial resolution could be used to obtain more detailed land covers, and I suppose that a greater detail in the land uses could link better with the acoustic signals. As the authors commented in the discussion (L352-353), not so high results “may be related to the use of a highly simplified land cover classification”
- With that broad land cover classification, it is necessary to analyze an image every week from early June and early September? are there so many changes in land uses during this period? I think there can be a lot of data redundancy in that analysis.
- In L177-180, the authors comment that “Imagery was postprocessed in Agisoft Photoscan software (version 2.7) and mosaicked as an orthophoto with three spectral bands (B, G, NIR). Orthophotos were georeferenced in QGIS Georeferencer using ground control points taken during the sample event and online imagery providers (Google earth).” If the final product of the Agisoft Photoscan software is an orthophoto, why the authors need to georeference this product? It has no sense. Additionally, do the authors consider Google Earth spatial accuracy enough to georefence a 2cm image? Revise bibliography: e.g. Zomrawi, Nagi & Ahmed, G. & Eldin, M.. (2013). Positional accuracy testing of Google Earth. Int. J. Multidiscip. Sci. Eng.. 4. 6-9.
- How the field data was collected? With a GPS?.
- The figure captions should be revised in the entire manuscript. Some of them explain more than expected in a figure caption (e.g. Fig.3 when the authors describe Plot a, it is not necessary to explain the type of curve and the meaning of certain values. This information should be in the text. Try to explain only what each figure is.
- Figure 2 is not referenced in the text.
- Be careful with the acronyms and revise the use of the acronyms throughout the text. For example, GLM is used 8 times but the authors do not explain what it is.
Author Response
Reviewer #1
Overall: I have reviewed the manuscript Land_779378. Knowing the manuscript from previous reviews in the Remote Sensing journal, I clearly recognize several improvements, but still think that the manuscript can be improved. Some of the comments and questions indicated in the previous review are not answered yet. Because pf this, I consider that there are some details that should be revised in order to improve the manuscript. My comments are detailed below:
Comment 1: The study compares two types of images with very high spatial and temporal resolutions but, are these high resolutions necessary? To classify herbaceous noncrop, herbaceous crop, woody (that merged to herbaceous noncrop to create a single noncrop vegetation category), built, and water, it is necessary a very high spatial resolution?. The very high spatial resolution could be used to obtain more detailed land covers, and I suppose that a greater detail in the land uses could link better with the acoustic signals. As the authors commented in the discussion (L352-353), not so high results “may be related to the use of a highly simplified land cover classification”
Response: These are good questions and we have adapted the text to try and be more clear. We propose the two spatial resolutions of imagery in order to resolve small habitat complexes embedded within our study areas. Narrow field margins for example, are identified by the UAS imagery, while the PS imagery (3 m) is not able to resolve these features. We comment on L63-65 whether different imaging systems alter capacity to observe non-agricultural habitats and feel this is a defensible rationale for testing a high resolution (1 m) and nearly high resolution (3 m) system.
One reason we kept a highly simplified land cover classification was that our primary objective was to identify a landscape/vocalizing bird richness relationship. Increasing the thematic resolution of the land cover classification complicated the analysis thus deviating from the central message. For this reason, we felt it was better left for future research.
We agree that these points are not clearly articulated within the text. We have supplemented L64-65 to read: ... to assess how different imaging systems might alter the capacity to observe non-agricultural habitats “which may be small or narrow” within intensively used agricultural landscapes… quotes only to emphasize added word: small.
We also have adapted L359-360 (with quotes to emphasize what was added):
This may also be related to the use of a highly simplified land cover classification “which was used to reduce complexity in the analysis, allowing our focus on the central objective of the study.”
Comment 2: With that broad land cover classification, it is necessary to analyze an image every week from early June and early September? are there so many changes in land uses during this period? I think there can be a lot of data redundancy in that analysis.
Response: We followed methods developed to map grassland characteristics which were cited. Herbaceous phenology is quite different over the time period used and while likely having some redundancy in the Blue and Green bands, will vary significantly in the Red and NIR. During editing versions we removed much of the introduction dealing with remote sensing rationale to improve clarity. We have adjusted the sentence (L210-212) referring to our approach to read as (quotes indicate changes):
SVM is a widely used classifier in grassland remote sensing because of its ability to detect variation of spectral reflectance in homogenous vegetation patches “with improved performance in grassland systems with increasing temporal resolution.”
Comment 3: In L177-180, the authors comment that “Imagery was postprocessed in Agisoft Photoscan software (version 2.7) and mosaicked as an orthophoto with three spectral bands (B, G, NIR). Orthophotos were georeferenced in QGIS Georeferencer using ground control points taken during the sample event and online imagery providers (Google earth).” If the final product of the Agisoft Photoscan software is an orthophoto, why the authors need to georeference this product? It has no sense. Additionally, do the authors consider Google Earth spatial accuracy enough to georefence a 2cm image? Revise bibliography: e.g. Zomrawi, Nagi & Ahmed, G. & Eldin, M.. (2013). Positional accuracy testing of Google Earth. Int. J. Multidiscip. Sci. Eng.. 4. 6-9.
Response: We agree that we can improve the description of our process here. Agisoft software combines image pixels using a coordinate system not associated with a projected coordinate system or datum, and so when it is exported, it is technically referred to as an orthophoto. We are updating the sentence to read:
A projected coordinate system (UTM 15N) was applied to the ungeoreferenced orthophoto using QGIS Georeferencer using ground control points taken during the UAS flight and an online map reference (Google earth) (Zomrawi et al. 2013).
We have also added the suggested citation to caveat our use of Google earth derived coordinates.
Comment 4: How the field data was collected? With a GPS?.
Response: The text was updated to be more specific. Field data was collected using an iOS smartphone application that collates geotagged photos and allows notes to be taken. Plant species observed were recorded.
I have updated the text (L202-204): “Ground truth data were collected using geo-tagged smartphone photographs and were accompanied by descriptions of plant species observed and then cross-referenced with an online map reference (Google earth) to guide delineation of training polygons for both UAS and PS imagery.”
I also moved the location of the sentence describing random pixel extraction to be directly after this section to improve clarity of the process.
L205-207: “Seventy percent of the pixels within the training polygons were randomly extracted and applied to the training algorithm, with the remaining 30 percent used to test model performance.”
Comment 5: The figure captions should be revised in the entire manuscript. Some of them explain more than expected in a figure caption (e.g. Fig.3 when the authors describe Plot a, it is not necessary to explain the type of curve and the meaning of certain values. This information should be in the text. Try to explain only what each figure is.
Response: We have revised the figure captions, removing unneeded explanations.
Comment 6: Figure 2 is not referenced in the text.
Response: We added a reference at L60.
Comment 7: Be careful with the acronyms and revise the use of the acronyms throughout the text. For example, GLM is used 8 times but the authors do not explain what it is.
Response: Revised.
Thanks for the excellent comments.
Reviewer 2 Report
The authors have addressed my comments on the previous version, and I have no further comments.
Author Response
Thank you for reviewing our manuscript.
Reviewer 3 Report
An excellent piece of research, congratulations! Well executed, flawlessly analysed, clearly delivered. I have only minor suggestions for edits. The only one of significance is to reframe the objective. You have it as “Our primary goal was to assess whether passive acoustic monitoring could be used to detect variations in avian biodiversity associated with differences in the composition and configuration of agricultural landscapes.” However, you have not measured the avian biodiversity as such but only one of its indicators. I suggest to express the objective similarly to how you framed your key result “measures of vocalizing bird diversity derived from acoustic recordings can be predicted from measures of agricultural landscape composition.” The more clearly stated objective would be “to assess whether vocalizing bird diversity derived from passive acoustic monitoring can be associated with differences in the composition and configuration of agricultural landscapes”.
And related to this important difference, I would like to see a comment on availability of evidence on the strength of correlation between the vocalizing bird diversity (from the acoustic recordings or surveys) and overall avian diversity in agricultural landscapes. Any references (see l. 371-372)?
My other comments are minor.
l 60 remove “Figure 1”
Broken Kettle Grasslands - Could be removed from the start as an obvious outlier (100% of noncrop vegetation) but it is really up to you. It seems to be redundant to mention it.
- 345-346: However, when comparing the much higher time and financial costs associated with UAS deployment and data processing in relation to PS imagery, this difference of 16% in explained variation seems less significant. – the actual estimates, if available, could be added to the Results
l 348-349: There were several UAS data points where vocalizing bird richness ranged from 4 to 14 while noncrop vegetation proportion remained approximately the same (about 25%; Figure 9a); a clear indication that unobserved factors may be influencing results. – you may need to describe the variation of noncrop vegetation types; or were these all only open margins, with no scrub/trees/ditches etc.?
- 371-372: we propose that it can serve as a useful and pragmatic metric for assessing avian diversity variations across agricultural landscapes that can be widely reproduced using inexpensive acoustic recordings. – any references to support this? Strength of correlation between the total avian species richness/diversity and measures of vocalizing bird diversity? Do you have any results for the costs of the method (incl. time analyzing the recordings) vs traditional bird counts?
Author Response
REVIEWER #3
Overall: An excellent piece of research, congratulations! Well executed, flawlessly analysed, clearly delivered. I have only minor suggestions for edits. The only one of significance is to reframe the objective. You have it as “Our primary goal was to assess whether passive acoustic monitoring could be used to detect variations in avian biodiversity associated with differences in the composition and configuration of agricultural landscapes.” However, you have not measured the avian biodiversity as such but only one of its indicators. I suggest to express the objective similarly to how you framed your key result “measures of vocalizing bird diversity derived from acoustic recordings can be predicted from measures of agricultural landscape composition.” The more clearly stated objective would be “to assess whether vocalizing bird diversity derived from passive acoustic monitoring can be associated with differences in the composition and configuration of agricultural landscapes”.
And related to this important difference, I would like to see a comment on availability of evidence on the strength of correlation between the vocalizing bird diversity (from the acoustic recordings or surveys) and overall avian diversity in agricultural landscapes. Any references (see l. 371-372)?
My other comments are minor.
Response: We have taken the suggestion to revise the objective statement (L60-62).
In regards to the second overall comment, the closest paper on the correlation between bird surveys and avian diversity in agricultural landscapes I have been able to find is the are the Best et al. 1983-1995 papers and the Lindsay et al. 2013 paper. We review this literature in L391-399. We also considered the North American Breeding Bird Survey, as well as eBird data but did not include these data sources because they are not suitable for comparison with the scale of analysis we used. Please see the further explanation on this in a later comment.
Comment 1: l 60 remove “Figure 1”
Response: Done.
Comment 2: Broken Kettle Grasslands - Could be removed from the start as an obvious outlier (100% of noncrop vegetation) but it is really up to you. It seems to be redundant to mention it.
Response: We decided to include since it is still situated among row crop agriculture, just not within 100 m.
We have updated the text (L300-302) to read: It is notable that BKG simultaneously had both 100% noncrop vegetation cover within 100 m of the acoustic recorder and the highest number of species identified at any site (22). We reported BKG results here since it was situated among row crop agriculture, just not within the 100 m radius of the acoustic recorder.
Comment 3: L345-346: However, when comparing the much higher time and financial costs associated with UAS deployment and data processing in relation to PS imagery, this difference of 16% in explained variation seems less significant. – the actual estimates, if available, could be added to the Results
Response: This is a fair point and would add value to the manuscript. We decided to add estimates of time required for the UAS image data collection and procession, as well as the cost in L348-353.
Comment 4: l 348-349: There were several UAS data points where vocalizing bird richness ranged from 4 to 14 while noncrop vegetation proportion remained approximately the same (about 25%; Figure 9a); a clear indication that unobserved factors may be influencing results. – you may need to describe the variation of noncrop vegetation types; or were these all only open margins, with no scrub/trees/ditches etc.?
Response: We felt this was a fair point and have included the names of the sites mentioned parenthetically to these lines in the text (L354) so that the viewer can compare with UAS imagery and LC in Figure 6. There were very few trees and shrubs and all were herbaceous margins.
Comment 5: 371-372: we propose that it can serve as a useful and pragmatic metric for assessing avian diversity variations across agricultural landscapes that can be widely reproduced using inexpensive acoustic recordings. – any references to support this? Strength of correlation between the total avian species richness/diversity and measures of vocalizing bird diversity? Do you have any results for the costs of the method (incl. time analyzing the recordings) vs traditional bird counts?
Response: We did not feel that the spatial extent of our study would be appropriately compared with other published measures of species richness. For example, the Breeding Bird Survey and eBird data would both include woodland and water birds which rarely occur in the cropping systems we analyzed. Our study site selection was partly guided by avoiding woodlands and water (L88-89).
We have added new text to L382-384 to indicate a need to verify our results with new research and explain why breeding bird survey and eBird are not comparable.
L471- “Additional studies will be needed to verify our results as they are not readily compared with existing datasets like the Breeding Bird Survey and eBird which operate at larger scales and include woodland and water birds that rarely occur in the cropping systems we analyzed”
We have also made a comparision in time and financial resources required for acoustic and point count surveys in L393-394.
Thanks for the excellent comments. They definitely improved the MS.
Reviewer 4 Report
The present study provides an interesting approach by linking remote sensing and passive acoustic techniques to understand how changes in agriculture cover explain bird species richness and soundscape indices.
A common concern when carrying out passive acoustic monitoring (PAM) is how sound propagates in the environment. Regarding the study sites, in general sound travels long distances in open areas such as agriculture areas, but the authors did a great effort to test for sound propagation in the study sites.
A few details should be explained or added.

Author Response
Overall: The present study provides an interesting approach by linking remote sensing and passive acoustic techniques to understand how changes in agriculture cover explain bird species richness and soundscape indices.
A common concern when carrying out passive acoustic monitoring (PAM) is how sound propagates in the environment. Regarding the study sites, in general sound travels long distances in open areas such as agriculture areas, but the authors did a great effort to test for sound propagation in the study sites.
A few details should be explained or added.
**PDF Comments and suggested amendments were included.
Responses:
PDF Comment L68: Study sites are the 100 m radius. Study areas were the different farms sites were located.
PDF Comment L90: This is a good point. We tested for distance between sites, and among North/South groupings of sites as we wanted to acknowledge the potential existence of regional patterns. We included these analyses in the supplementary materials. Results suggested that sites that were clustered where not impacting the results holding the other variables constant.
PDF Comment L92: We placed the parameters used to set up the recorders is detailed in L100-101.
PDF Comment L152: We placed the rationale for recording 1 min every 10 minutes is detailed in L100-101.
PDF Comment L197: This was a good catch and the reader will appreciate our inclusion of these. We added a brief description of all land cover classes.
Thanks for the comments.
Reviewer 5 Report
This is an interesting research showing the potential usefulness of linking high-resolution remote sensing with passive acoustic monitoring of birds to provide a useful biodiversity indicator in intensively managed agricultural landscapes.
Although the manuscript is well written, I would like to see clarification of the limitations of the survey with regard to the fact that the bird surveys were only conducted in the late breeding season of one year and the sampling dates did not coincide between survey sites. I also found that the species accumulation curves did not reach an asymptote in some sites, as shown in Figure 5, indicating that the sampling of birds was incomplete (some species were not detected). As such, I disagree with the statement that “both sampling approaches demonstrated adequate sampling saturation” (supplementary materials). This issue would be particularly important if such undetected species include species of conservation concern. Note, I don’t deny the use of species accumulation curves; even if the curves do not reach an asymptote, species richness estimators can be used to compare the richness of collections and to assess the stability of the biodiversity estimates with increasing sampling.
Specific comments:
- P3L92-101, While not required, it would be easier for many readers to understand the methods if there was a picture showing the installation of acoustic recorders as a supplementary material.
- P8L268-271, The Dawn recording method (a) -> (b)?
the Bioacoustic method is (b) -> (a)?
- Figures 7,8, Please clarify that black lines indicate the estimated coefficients, and grey polygons indicate their 95 % confidence intervals.
Author Response
Overall: This is an interesting research showing the potential usefulness of linking high-resolution remote sensing with passive acoustic monitoring of birds to provide a useful biodiversity indicator in intensively managed agricultural landscapes.
Although the manuscript is well written, I would like to see clarification of the limitations of the survey with regard to the fact that the bird surveys were only conducted in the late breeding season of one year and the sampling dates did not coincide between survey sites. I also found that the species accumulation curves did not reach an asymptote in some sites, as shown in Figure 5, indicating that the sampling of birds was incomplete (some species were not detected). As such, I disagree with the statement that “both sampling approaches demonstrated adequate sampling saturation” (supplementary materials). This issue would be particularly important if such undetected species include species of conservation concern. Note, I don’t deny the use of species accumulation curves; even if the curves do not reach an asymptote, species richness estimators can be used to compare the richness of collections and to assess the stability of the biodiversity estimates with increasing sampling.
Response: We agree that clarification of limitation of the survey in regards to time of year and single season are needed in the discussion. In our paragraph about geographical limitations of the study, we have added a few points of discussion regarding these points.
In L374-378 we added: “Another important consideration is that our study occurred late in the summer and in a single season. While we would expect the same community of resident breeding birds throughout the growing season, we acknowledge that results may have been different if acoustic activity was sampled a few months earlier when birds are establishing territory and pair-bonding. We look forward to future studies using long duration passive acoustic monitoring across multiple seasons which are able to determine the precise effects of seasonality and species detection.”
Response: The claim that we achieved species saturation in the SACs was dialed back. This was a good point may have gone too far. The phrase has been revised to read:
“Both sampling approaches demonstrated trends towards reaching sampling saturation”
Comment 1: P3L92-101, While not required, it would be easier for many readers to understand the methods if there was a picture showing the installation of acoustic recorders as a supplementary material.
Response: We have placed a figure with several instances of acoustic recorder installation in the supplementary materials.
Comment 2: P8L268-271, The Dawn recording method (a) -> (b)?
the Bioacoustic method is (b) -> (a)?
Response: This was a great catch. Thanks for pointing out.
Comment 3: Figures 7,8, Please clarify that black lines indicate the estimated coefficients, and grey polygons indicate their 95 % confidence intervals.
Response: Completed for Figure 8 (a & b) and 9.
Thanks for the excellent comments.